# Sourdough Microbiome Comparison and Benefits

**DOI:** 10.3390/microorganisms9071355

**Published:** 2021-06-23

**Authors:** Siew Wen Lau, Ann Qi Chong, Nyuk Ling Chin, Rosnita A. Talib, Roseliza Kadir Basha

**Affiliations:** Department of Process and Food Engineering, Faculty of Engineering, Universiti Putra Malaysia, Serdang 43400, Selangor, Malaysia; raclsw96@gmail.com (S.W.L.); angieannqi@gmail.com (A.Q.C.); rosnita@upm.edu.my (R.A.T.); roseliza@upm.edu.my (R.K.B.)

**Keywords:** sourdough, fermentation, microbiota, benefits, health

## Abstract

Sourdough is the oldest form of leavened bread used as early as 2000 BC by the ancient Egyptians. It may have been discovered by accident when wild yeast drifted into dough that had been left out resulting in fermentation of good microorganisms, which made bread with better flavour and texture. The discovery was continued where sourdough was produced as a means of reducing wastage with little known (at that point of time) beneficial effects to health. With the progress and advent of science and technology in nutrition, sourdough fermentation is now known to possess many desirable attributes in terms of health benefits. It has become the focus of attention and practice in modern healthy eating lifestyles when linked to the secret of good health. The sourdough starter is an excellent habitat where natural and wild yeast plus beneficial bacteria grow by ingesting only water and flour. As each sourdough starter is unique, with different activities, populations and interactions of yeast and bacteria due to different ingredients, environment, fermentation time and its carbohydrate fermentation pattern, there is no exact elucidation on the complete make-up of the sourdough microbiome. Some lactic acid bacteria (LAB) strains that are part of the sourdough starter are considered as probiotics which have great potential for improving gastrointestinal health. Hence, from a wide literature surveyed, this paper gives an overview of microbial communities found in different sourdough starters. This review also provides a systematic analysis that identifies, categorises and compares these microbes in the effort of linking them to specific functions, particularly to unlock their health benefits.

## 1. Introduction

The use of sourdough as a means of leavening is one of the ancient methods of grain fermentation [1]. Grinding of cereals, pseudo-cereals or legumes followed by addition of water brings about the formation of dough, which subsequently turns into sourdough after a period of time [2]. The presence of a symbiotic colony of lactic acid bacteria (LAB) and yeasts inhabiting our diverse ecosystem induce lactic acid fermentation of sourdough which eventually becomes a stable culture after many hours [3]. The culture can be sourced from three established methods, resulted in classification of sourdough into types I, II and III. Type I sourdough refers to the traditional sourdough that requires uninterrupted propagation (backslopping) by refreshing using fresh flour and water at regular intervals [4]; type II sourdough inoculates adapted cultures industrially as dough acidifiers; whereas type III sourdough is usually dried for easy storage and utilisation [5,6,7]. Type I sourdough can be categorised as type Ia, containing pure culture sourdough starters with different origin; type Ib, being refreshed everyday and fermented spontaneously; and type Ic, which its origin is tropical countries with higher fermentation temperature [8,9]. At the same time, some researchers also considered another type of sourdough as type 0, referring to pre-doughs or sponge doughs, with addition of baker’s yeast (*Saccharomyces cerevisiae*) [5]. To ensure the manufacture of products being optimised and consistent, manufacturers are always keen to identify and develop sourdough types II and III [10]. Despite being relatively unstable, costly and time consuming, the type I sourdough is more commonly utilised for microbiome study due to its natural diversity [11].

Scientists have started to discover that the variability in number and type of microbiota in dough depends not just on the native microbial flora of the baker’s environment and hands, but also other factors like choice of flour, when and how often the starter is fed, dough hydration level and type of cereal used, leavening temperature, fermentation time and sourdough maintenance temperature [12,13,14]. Different metabolic pathways in sourdough add flavourful metabolic by-products to the mixture, which result in subtler but more complex flavor [15]. Each microbial community has the ability to produce unique flavour profile as some yield yoghurt-like flavour from lactic acid while others produce sharper, more vinegary note with acetic acid (https://truesourdough.com/18-ways-to-make-sourdough-bread-more-or-less-sour/, accessed 21 March 2021) [16]. Lactic acid, together with vinegary acetic acid, contributes to sourdough’s tangy characteristic. Hence, with the aid of sourdough, dissatisfying sensory qualities especially the flavour and mouth-feel of some gluten-free bakery products can be enhanced [17,18], thus boosting their palatability and market values.

As microbial processes show significant potential in improving organoleptic characteristics and design of nutritional quality and health effects of foods and ingredients, sourdough uses have spread widely to other food products i.e., crackers, waffles, pancakes, tortillas, muffin and noodles beyond breads [19]. Bread, being one of the basic staple foods in many countries and designated the ‘staff of life’ is believed to be the prior option of sourdough addition from earlier days [20]. The sourdough bread trend was not a major highlight until recently when it re-emerged, and people have become enthused about home-baked sourdough breads. In 2020, there was a boom in the sourdough bread trend as millions are confined to their homes during the Covid-19 pandemic. The sourdough bread revival has gone into overdrive and was on third Google recipe search globally (https://trends.google.com/trends/yis/2020/GLOBAL/, accessed 21 March 2021) [21]. The underlining reasons that have been spurring sourdough bread boom include various health benefits which makes the discussion in this paper timely. The different factors affecting sourdough fermentation and their relationships to the benefits of sourdough intake due to the presence of beneficial microorganisms are also included.

Figure 1 shows a mindmap guide to sourdough. Sourdough is the result of fermentation from two basic ingredients, mainly flour and water. Microbiome in sourdough is not only affected by various components in different types of flours but also dough hydration level, backslopping time, fermentation time and temperature. Understanding the factors affecting a microbiome enables a greater accrued understanding of microbial ecology, which allows a better selection of starter cultures and fermentation conditions. Microbial succession is also mentioned, which involves the inheritance from ingredients, adaption of acidic conditions and domination by acid-tolerant sourdough bacteria. Three types of carbohydrate fermentation patterns of bacteria are included in sourdough which are obligately homofermentative, obligately heterofermentative and facultatively heterofermentative. With the presence of beneficial microorganisms, sourdough stands out to bring various benefits to properties of bread and health which are also listed in Table 1.

## 2. Microbial Communities in Different Sourdough Starters

Sourdough starter, which is also known as natural yeast, has conducive habitat for microbes that can support growths of more than 50 species of lactic acid bacteria (LAB) and more than 20 species of yeasts [8]. As technology advances, many new species of LAB are discovered. For instance, in March 2020, there are already 262 species being reported under the genus *Lactobacillus* [54]. Most recently, in study of Zheng et al. [54], the genus *Lactobacillus* was proposed to be reclassified into 25 genera after reviewing the taxonomy of Lactobacillaceae and Leuconostocaceae. Consequently, some names of LAB were updated. They included *Fructilactobacillus sanfranciscensis* (formerly known as *Lactobacillus sanfranciscensis*), *Lactiplantibacillus plantarum* subsp. *plantarum* (formerly known as *Lactobacillus plantarum* subsp. *plantarum*) and *Levilactobacillus brevis* (formerly known as *Lactobacillus brevis*) [54]. For this review, the old names are referred to for reasons of familiarity amongst researchers due to this very new discovery.

The different species require different conditions for their optimal growth, hence the proportion of each species found in each sourdough starter also varies. Towards the final stage of fermentation, sourdough starters are asserted to be mature when the cell densities and abundances for all LAB and yeasts reach a plateau [55,56]. The species that are more competitive and adaptive in mature sourdough are usually those with larger population [57]. The divergence of microbial profile from one sourdough starter to another can also be observed when the types of ingredients (mainly flour and water), number of sourdough propagation steps, fermentation time, fermentation temperature, dough hydration, leavening temperature and baking environments are different [1,11,14,58,59,60,61].

### 2.1. Ingredients as Sources of Microorganisms

Without any prior sterilisation, flour is milled from raw cereals which are naturally inhabited by certain types of microorganisms. Corsetti et al. [62] isolated LAB from wheat and some non-conventional flours originated in Italy and revealed the harbouring of mainly lactobacilli and enterococci in the cereals. De Angelis et al. [63]’s list of microbes included *Acinetobacter*, *Pantoea*, *Pseudomonas*, *Comamonas*, *Erwinia*, *Sphingomonas* and *Enterobacter* from natural wheat (soft or durum) flour from Southern Italy. These bacterial populations are believed to be pioneers of sourdough microbiota since the lowest dissimilarity was found between microbial communities in flour and sourdough starters, as compared to those from water source, bakers’ hands and dust [58]. In the study of Rizzello et al. [64], all sourdough starters were harboured with *Leuconostoc citreum*, *Lactobacillus plantarum* and *Lactococcus lactis*, which are almost similar to the microbial profiles of dough prior to fermentation. These facts hence supported the assertion that the types of flour chosen for making sourdough affect the microbial communities remarkably. Some other factors affecting sourdough microbiota, including fermentation temperature and backslopping time, are also closely related to the types of flour used since microbes present in flour naturally prefers different fermentation conditions [65]. The types of microbes which grow in sourdough starters from different batches or types of flour are not necessarily the same due to the different quantities and qualities of carbohydrates, proteins, minerals, lipids and enzyme activities [11], different amount of microbial growth factors, and also due to the presence of microbial growth inhibitors [57].

The distinction of microbial profiles found in sourdough starters can be possibly seen by comparing the farming conditions (climatic conditions, farming systems, parasitic or fungal attack, etc.) and the degree of milling of flours [62,63]. Studies that directly relate farming systems and sourdough microbiome are still limited. Rizzello et al. [64] claimed that if cereals used for making sourdough are grown conventionally, the sourdough starter usually has lower biodiversity than those using flour from organically grown cereals. The study of Gobbetti et al. [66] then complemented the claim by limiting it within the Firmicutes phylum. For example, sourdough starter using conventionally grown flour only consisted of *Leuconostoc*, *Lactococcus*, and *Lactobacillus*, whereas in organically grown flour, other LAB such as *Pediococcus* were also found [64]. Adding on, during the milling process of grounding cereals into flour, some essential nutrients for sourdough LAB, particularly protein and ash, are being stripped together with the germs [67]. The degree of milling (amount of bran left on milled grains) is represented by the flour extraction rate, whereby a higher extraction rate is equivalent to a lower degree of milling [63]. By using flour with higher extraction rate, lactic acid fermentation can carry on for longer time with improved microbial growth and acidification power despite the low pH condition in sourdough starters [1,68]. This is due to the raised buffering capacity and contents of minerals and micronutrients in flour with higher extraction rate [69,70]. Hence, the sourdough environment for microbial activities is claimed to be more sustainable, leading to the release of a greater amount of lactic acid without affecting the final pH [68]. The sustainable sourdough environment also allows higher microbial diversities in sourdough starters using flour with higher extraction rate [63]. Generally, rye flour has higher extraction rate than wheat flour [71], thus the greater amount of protein and ash brings about a much wider variety of microbes in rye sourdough as compared to wheat sourdough.

Table 2 and Table 3, respectively, summarise some of the LAB and yeasts found in cereal dough before and after fermentation following the types of cereals, including wheat, rye, spelt, chickpea, etc. Among the flours, wheat and rye are most studied and mentioned. Naturally, both the flours possess different percentages of life-supporting components for LAB and yeasts, such as fermentable sugars, pentosans and amylase enzyme. Rye has a higher amount of amylase than wheat, thus boosting amount of readily fermentable sugars for LAB in a shorter period of time (http://www.baystatemilling.com/ingredients/why-not-rye/, accessed 21 March 2021) [72]. Fujimoto et al. [73] realised a greater rise in Gram-negative bacterial count together with reduction of sugar content on the first day of fermentation for rye sourdough due to higher enzymatic activity as compared to wheat sourdough. The subsequent content of organic acids was affected too. Fraberger et al. [74] who studied sourdough in Austria isolated only *Leuconostoc* spp. and streptococci from wheat sourdough, whereas some exclusive LAB such as *Lb. diolivorans*, *Lb. gallinarum*, *Lb. kimchii*, *Lb. otakiensis*, *Lb. parabrevis*, *Lb. paralimentarius*, and *Lb. xiangfangensis* were found in rye sourdough. Despite the different composition of every flour, *Lb. plantarum* is one of the frequently identified LAB [10] as it is found in wheat [75], spelt [76], rye [77] as well as sourdough utilising composite flour of wheat and legumes [78]. The other common sourdough LAB is *Lb. sanfranciscensis* and it has a high fitness in conditions of natural sourdoughs [13,64,78,79,80].

The utilisation of different types of flour during fermentation of sourdough also leads to the diversity of wild yeast species, for example, *Saccharomyces cerevisiae*, *Candida holmii* [2], *C. humilis* and *Kazachstania exigua* [3]. Unlike LAB, the microbial profile for yeast is more homogeneous as there are fewer yeast species [93]. Among them, *S. cerevisiae*, a facultative anaerobe that metabolises glucose with and without oxygen [94] is most often isolated from several studies of sourdough [10] which included the wheat sourdough [14,56,64,74,79,95], rye sourdough [56,74,84,85], sorghum sourdough [59,89], corn sourdough [86], rice sourdough [88] and as well as sourdough from composite flour (wheat, rye and grapes) [92].

Although cereals are typically chosen for ingredients of sourdough, some bakers prefer the non-conventional routes. Since the substrates in non-conventionally used flours are different in terms of composition and forms, the fermented sourdough products possess a more diverse and complex mixture of LAB, and that include their metabolites [97]. For instance, chickpea sourdough was studied by Corsetti et al. [62] and Galli et al. [98], while Jagelaviciute and Cizeikiene [97] claimed that quinoa, hemp and chia flours are qualified ingredients for sourdough fermentation as *Lb. sanfranciscensis* with sufficiently high final LAB counts were detected. On the other hand, unlike cereals, some ingredients like fermented apple juice, yoghurt and grape may contain insufficient substrates to sustain self-fermentation. Therefore, they are often incorporated into sourdough as an additional inoculum to form composite sourdough and have shown an enhanced microbial diversity [91,96]. *Lb. paracasei*, as the dominant LAB, had inhabited spontaneous Brazilian grape composite sourdough from wheat and rye [92], while *Lb. plantarum* gave the most desirable characteristics in the composite sourdough involving cassava, sweet potato, and soybean [99].

Apart from the flour, water is also a prerequisite for making sourdough. In contrast to flour, water has not been included as one of the significant factors in affecting the sourdough microbiome despite its role as the key ingredient for making sourdough other than flour [66]. Minervini et al. [100] proposed some significant relationship between water hardness and spontaneous sourdough microbiota. The concentration of potassium ions in tap water used during fermentation period and relative abundances of *Lb. plantarum* strains (r  =  0.86) and *W. confusa* (r  =  0.80) were positively correlated. There was also a positive correlation concluded between the composition of sulphate ions in water and the number of *P. pentosaceus* (r  =  0.82). The results of Minervini et al. [100] above suggested that water gave significant effect on the development of sourdough fermentation. However, they found that for mature sourdough, the impact of water on the sourdough microbiota is less obvious. At this juncture, minerals found in water used for sourdough starter fermentation may be considered as a booster for microbe growth during its early phase of fermentation development as some effects were shown.

### 2.2. Microbial Succession along Fermentation Period

A sourdough colonised by various LAB and yeasts is an energising and complex ecosystem, which constantly evolves since the first mix of water and flour. The development of microbial composition within sourdough which includes the adaptations and interactions among microbes can be studied as well as compared through manipulation of technical, environmental and ingredient parameters [56]. Often, the optimum length of fermentation time is targeted in grasping the best outcome. However, Oshiro et al. [82] summarised some of the obstacles, for instance, the uncontrollable succession of LAB due to spontaneous fermentation as well as the limitations of technology in quantifying them precisely and quickly. Despite that, the succession of microbes in sourdough starters during the fermentation period can still be described generally. The process of one species successively giving way to another species until the maturation of sourdough is divided into three phases [65,82,87]. Most studies tracked the microbial dynamics for ten days, for example, Ercolini et al. [56] detected the high abundance of microbes which were naturally inherited from cereals [101], such as *Acinetobacter*, *Pseudomonas*, *Sphingomonas* and especially *Enterobacter*, during the first phase, followed by domination of *Weissella* spp. and *Lb. lactis* in the second phase (day 2 to day 5), and lastly, after day 5 (third phase), the *Leuconostoc* spp. and the *Lb. sakei* were those prevailing in the microbiota of wheat sourdough. Similarly, Van der Meulen et al. [76] studied the microbiota changes daily whereby those species that are not specific for sourdough include *Enterococcus*, *Lactococcus* and *Leuconostoc*, dominated the first phase of microbial succession. Then, the percentages of sourdough LAB, like *Lactobacillus*, *Pediococcus*, and *Weissella*, surpassed the others in the second phase and finally only well-adapted LAB *(Lb. plantarum and Lb. fermentum*) were able to dominate in the sourdough ecosystem.

Corsetti and Settanni [102] also deduced that LAB cocci, especially *P. pentosaceus*, had significant abundances during early phases of sourdough fermentation but not in mature sourdough. After all, the microbial succession phases before reaching mature stage of sourdough are natural selection processes [61], where *Lactobacillus* spp. that can tolerate and adapt to extreme acidity, temperature, dehydration or other specific conditions better then dominate the ecosystem [79]. Under some circumstances, other sourdough bacteria like the *P. pentosaceus* and *L. citreum* which have greater growth abilities or faster adaptation in sourdough have also shown potential to perform better than the lactobacilli [61].

The microbial profiles, acidifying and leavening properties may not converge to similar values for all mature sourdough at the end of fermentation as they vary from one type of flour to another [56]. Along the fermentation time, sourdough made from rye, wheat, spelt and some untypical ingredients like faba bean, usually reach maturity during days 5 to 7 [56,76,90]. Weckx et al. [83] who studied rye sourdough reported that it took shorter time to reach maturity as compared to wheat and spelt sourdough, possibly due to the different rates of enzymatic activities. In mature sourdough, a range between 10^7^ and 10^8^ colony-forming units (cfu) g^−1^ [103] or sometimes more than 10^8^ cfu g^−1^ [4,56,87] of LAB can be found. Therefore, it can be said that LAB population increases as pH reduces and finally both reach a plateau when sourdough mature [12]. LAB are the dominant microbes found in sourdough and typically have a LAB to yeast ratio of 100:1 [56,95,104]. Yeast is usually found to be within the range of 10^6^ to 10^7^ cfu g^−1^ in mature sourdough [3,14].

### 2.3. Carbohydrate Fermentation Patterns of Lactic Acid Bacteria (LAB)

Generally, most microorganisms especially the LAB found in sourdough starters that ingest carbohydrates (around 75% of flour by weight) [94] in cereal fermentation originated from the genus *Lactobacillus*, such as *Lb. sanfranciscensis*, *Lb. plantarum* and *Lb. helveticus* [1,61,102]. However, the kinetics of this biochemical reaction for *Lactobacillus* species is not entirely the same. To ease the understanding these bacteria’s behaviour, they are well classified into three groups, i.e., obligately homofermentative, obligately heterofermentative and facultatively heterofermentative, following their carbohydrate fermentation patterns as listed in Table 4.

Carbohydrate metabolisation is the most essential process during sourdough fermentation [48]. During this process, the LAB consume simple sugars, the building blocks of carbohydrates as substrates continuously for the generation of life-sustaining energy. The microbes then convert and release carbohydrates in the form of organic acids (usually lactic acid and acetic acid), carbon dioxide and/or ethanol [48]. The homofermentative LAB produce at least 90% of lactic acid and only little amount of acetic acid, whereas heterofermentative LAB produce a more balanced proportion of lactic and acetic acids, as well as other organic compounds such as carbon dioxide, ethanol and acetaldehyde [1,71,94]. The difference between the obligately and facultatively heterofermentative LAB is that besides producing equimolar of lactic acid and acetic acid, the latter can produce carbon dioxide as one of the by-products without the presence of glucose but by using gluconate [1]. In real situations, heterofermentative LAB are more common and significant in sourdough fermentation. Table 5 compares the different characteristics of LAB from the three carbohydrate fermentation patterns. Almost all (at least 95%) sourdoughs studied in Europe, United States and Canada were found to be dominated by solely heterofermentative LAB or both the heterofermentative with homofermentative lactobacilli simultaneously together [107].

With knowledge on the different characteristics of LAB resulting from different carbohydrate fermentation patterns, Axelsson and Ahrné [108] proposed one easy method to determine the group of LAB in sourdough, i.e., through carbon dioxide detection. If carbon dioxide is absent, the sourdough can be said to contain only homofermentative LAB as they produce almost solely lactic acid through glycolysis, whereas heterofermentative LAB release carbon dioxide, organic acids including lactic and acetic acids, and sometimes ethanol. The homofermentative LAB in the sourdough only consume hexoses, while pentoses are consumed by the heterofermentative LAB (both obligately and facultatively) which involve the phosphoketolase enzyme. Aided by the enzyme fructose-1,6-diphosphate aldolase, facultatively heterofermentative LAB sometimes fall under the homofermentative group as they also digest carbohydrates using the same pathway, i.e., Embden–Meyerhof–Parnas (EMP) as obligately homofermentative LAB, instead of phosphoketolase pathway for obligately heterofermentative LAB [1,71,105]. Both the obligately homofermentative and facultatively heterofermentative LAB dominate the sourdough microbiota when fermentation temperature is above 30 °C, but when the sourdough starter is kept at temperature below 30 °C, it comprises primarily obligately heterofermentative LAB [93].

Arising from their differences in carbohydrate fermentation patterns, sourdough products possess their distinctive characteristics too. For example, Hansen and Hansen [109] studied one of the heterofermentative LAB, *Lb. sanfranciscensis* which gave sourdough bread more soft and acceptable sour taste than another homofermentative LAB, *Lb. plantarum*, due to higher acetic acid content that probably enhanced other flavour compounds. Moreover, the rye dough and bread containing heterofermentative LAB, *Lb. brevis*, showed lower pH and higher organic acid production compared to those with homofermentative LAB, *Lb. plantarum* [75]. From rheological aspect, homofermentative LAB, with addition of yeast, also contributed to dough softening [75]. Hence, both the heterofermentative and homofermentative LAB play different roles in optimising and maintaining qualities of sourdough products.

## 3. Benefits of Sourdough

Sourdough is considered a gift of bread because it provides a wealth of benefits. Application of sourdough has many advantages towards enhancing the quality of food products including shelf life, aroma, texture and nutritional value [12,18,88,110]. Properties of bread is improved as sourdough has better protection against spoilage [111] and has extended shelf life due to its acidic environment containing organic acids like acetic, caproic, butyric and propionic acids which reduces development of deleterious microbes [22]. The release of organic acids and pH reduction in sourdough bread by *Lb. sanfranciscensis* CB1 can stop the growth of contaminating or spoiling flora including *Aspergillus*, *Fusarium*, *Monilia* and *Penicillium* [85,112]. Enhanced antifungal action was found with increased microbial complexity of sourdough. The best result was achieved with microbial combinations more than 5 LAB isolates [10]. Jenkins [113], Torrieri et al. [114] and Fernández-Peláez et al. [115] have proposed that prolonged shelf life and delayed staling of sourdough bread are contributed by the decomposition of starch during lactic acid fermentation.

In addition, the fermentation process of sourdough is highlighted to bring more flavour, complexity, aroma and better texture of bread due to LAB enzymatic hydrolysis processes and the generation of compounds in the Maillard reaction during the baking process [23,24]. The formation of flavour volatiles in sourdough is contributed by proteolytic processes of amino acids, conversion of glutamine released from cereal proteins to glutamate, and conversion of arginine to ornithine by LAB during baking which have a profound influence in a more tangy and roasty sourdough taste. The crumb aroma, flavour and texture are obtained by microbial and enzymatic reactions such as lipid oxidation by cereal enzymes, Ehrlich degradation of amino acids and formation of metabolites from LAB [116,117,118,119,120,121]. Lipid oxidation is usually initiated with the presence of active enzymes such as lipoxygenase in raw material followed by oxidation of polyunsaturated fatty acids into free radicals, peroxides and hydroperoxides which will be converted into volatile compounds during baking [25]. Based on the recipe and process used, the oxidation of lipids in sourdough and bread induce the formation of different aldehydes, alcohols and esters which influence the aroma and flavour of bread crumb. In addition, LAB and yeast in sourdough produce aroma precursors such as free amino acids, which lead to the generation of aldehydes or corresponding alcohols by Ehrlich pathway [119]. 3-Methyl-1-butanol is probably the most crucial fermentation aroma compound in bread crumb which originate from degradation of amino acid [122]. The fermentation process gives rise to volatile profile modification and confer a specific aroma in sourdough and bread crumbs [25]. It is shown that sourdough bread has a greater content of volatiles and achieves a higher score in sensory evaluation compared to bread chemically acidified with lactic and acetic acids [119]. Acetic acid (at levels of 100–200 ppm) formed during sourdough fermentation was found to act as a flavour enhancer in the bread crumb [109]. In terms of texture, a positive impact was shown with presence of metabolites such as exopolysaccharides produced by LAB in sourdough [123]. Exopolysaccharides has a great water retention (due to polysaccharide-water binding ability), which in turn appears in reducing crumb hardness [123,124,125]. Exopolysaccharides, especially β-glucans [126], also contribute to the viscoelastic properties of dough [48], leading to increased volume of bread [11]. Bockwoldt et al. [126] found out that sourdough with *P. claussenii* achieved higher maximum concentration of β-glucans within shorter time than sourdough with *Lb. brevis.* The β-glucans production rate was mostly affected by fermentation temperature, where 25 °C was more preferred by *P. claussenii* and 35 °C was more conducive for *Lb. brevis* [126]. Recently, sourdough has been used in the baking industry to produce breads with reduced usage of additives (‘label cleaning’) [127], improved organoleptic characteristics and nutritional content [29,128]. Most of the off-the-shelf breads undergo modifications in production technology and addition of additives such as enzymes, preservatives, emulsifiers and improvers to speed up baking process and meet consumer acceptance in maintaining freshness and extending shelf life of bread [129]. A lot of these additives are to be blamed for inducing symptoms related to gastrointestinal disorder in people. Although sourdough bread is likely to have similar constitution as sourdough bread has the exact same flour as conventional bread, its nutritional content however is comparatively higher [32]. Traditional sourdough bread is also a healthier option for people who have intolerance towards commercial baker’s yeast in conventional breads [32].

Sourdough also provides many health benefits, particularly to the gastrointestinal system. The natural activities of LAB in sourdough ingesting carbohydrates in flour, diminish the contents of non-digestible oligosaccharides, fructans and raffinose (known as the FODMAPs) which are poorly assimilated in our body [26]. FODMAP stands for fermentable oligosaccharides, disaccharides, monosaccharides and polyols, which are short-chain carbohydrates (sugars) that the small intestine absorbs poorly. Some people experience digestive distress after eating them. During fermentation, sourdough microorganisms, especially yeast *S. cerevisiae* is responsible for the production of enzyme invertase which permits the declination of 50–80% of fructans levels in flour or wholemeal in bread production [27]. Struyf et al. [27] have developed a yeast-based strategy using co-cultures of *S. cerevisiae* and *Kluyveromyces marxianus* to reduce FODMAP levels in bread. The alteration in carbohydrate content due to degradation of oligosaccharides by sourdough LAB gives more toleration to gastrointestinal patients [130]. As wheat grains have a high level of fructan of approximately 0.9–2.7% dm [131], sourdough bread is preferred by patients with gastrointestinal disorders such as irritable bowel syndrome (IBS). IBS is a common disorder of the gut that can cause host negative symptoms affecting the intestine and resulting in chronic or recurrent abdominal pain and altered bowel patterns [132,133].

In addition, risk of gluten in bread for people suffering from gluten-related disorders can be eliminated with selected sourdough lactobacilli, including *Lb. sanfranciscensis*, *Lb. rossiae*, *Lb. plantarum*, *Lb. brevis*, *Lb. pentosus*, *Lb. alimentarius*, *Lb. fermentum*, *Lb. paracasei*, *Lb. casei* subsp. *casei*, and *P. pentosaceus* [28]. Although gluten is responsible for the baking properties of wheat [134], it leads to the production of gluten proteins, namely gliadins (prolamins) and glutenins (glutelins) that trigger the immune-mediated adverse responses in our body, causing inflammation and damage to the small intestine of celiac disease patients [135,136,137]. Celiac disease is known as a gluten intolerance syndrome [137,138,139] or gluten-sensitive enteropathy, a genetically-determined chronic inflammatory disease of small bowel against undigested peptides derived from dietary gluten in wheat, rye and barley [135,140,141,142,143]. There are two features of celiac disease pathogenesis whereby one is the inability to digest proline-rich gluten polypepties to generate peptides smaller than nine amino acids while the other is the deamidation of glutamine residues of such peptides by tissue transglutaminase [144]. Sourdough fermentation has been reviewed for modification of immunogenic sequences (epitopes) of gluten proteins for dietary treatment of celiac disease [144,145]. These include long-time sourdough fermentation and enzymatic modification so that epitopes are no longer recognized by immune system of celiac disease patients [144]. For instance, sourdough fermentation disrupts the network of gluten protein. Glutenins, the highest molecular weight proteins in gluten are polymers stabilized by disulphide bonds. When glutenins are partially hydrolysed during sourdough fermentation, depolymerisation and solubilisation of polymers happen [146]. Additionally, when pH is slightly acidic during the first hour of sourdough fermentation, glutathione (an endogenous reducing agent in dough) allows cleavage of disulphide bonds particularly [144]. At last, disruption of the gluten network releases proline-rich polypeptides which will be exposed to the action of proline-specific peptidases from lactobacilli in sourdough [144]. In recent research [147], gliadin degradation with *Lb. paracasei* LPA4 was found to exhibit the strongest alterations of the gliadin pattern, followed by *Lb. plantarum* LPl5 among the tested LAB.

Besides gluten, the presence of phytic acid which leads to gastrointestinal disorder, digestive discomfort and flatulence (https://sustainablefoodtrust.org/articles/sourdough-and-digestibility/, accessed 21 March 2021) [33,148] can also be diminished with sourdough fermentation. The presence of a high amount of phytic acid in bread interferes with digestive enzymes like the trypsin, pepsin and amylase and reduces their capability to break down proteins, starches and fats in the stomach acid. (https://chriskresser.com/another-reason-you-shouldnt-go-nuts-on-nuts/, accessed 21 March 2021) [149,150]. During fermentation, the LAB and yeast in sourdough, i.e., *Lb. brevis*, *S. diasticus*, *S. cerevisiae* and *Lb. fermentum* produce the enzyme phytase which neutralise phytic acid thus free up these digestive enzyme to allow more effective digestion [29,30,31]. A recent screening of 152 LAB isolated from cereal-based substrates revealed a widespread capacity of the isolates (95%) for degrading phytic acid with strains of *Lb. brevis* LD65 and *Lb. plantarum* PB241 showing the highest phytase activity while *W. confusa* strains showed low or no phytase activity [151]. Sourdough bread is claimed to improve nutrition via pre-digestion of phytic acid and grains with the presence of phytase produced by LAB as it makes food more easily digestible than yeast-fermented or non-fermented breads (https://sustainablefoodtrust.org/articles/sourdough-and-digestibility/, accessed 21 March 2021) [32,33]. Besides phytase, the presence of lactobacilli itself in fermented sourdough contains probiotics and prebiotics which aid in digestion [34,35,36]. A probiotic is defined as viable bacteria that have a beneficial effect of improving or restoring the gut flora when ingested. Although baked sourdough bread may not contain probiotics as they are not able to survive the high heat during baking process, they still, however, contain prebiotics. Prebiotics are food or compounds in food which are beneficial to the gut bacteria and confer health benefits and well-being upon their host [34,152]. One of the potential prebiotics released by LAB in sourdough starters is β-glucan, shown by Pérez-Ramos et al. [37], which boosted probiotic (*Lb. plantarum* WCFS1) viability in both oat and rice sourdough, though under starving conditions. Both probiotics and prebiotics share a crucial role in the manipulation of bacterial activities that colonise the digestive system in our body.

Besides supporting the gut microbiome, the consumption of probiotics and prebiotics in adequate amount helps in controlling glycaemic index (GI). GI is the speed at which sugar enters our body’s bloodstream [35,153]. High GI causes blood sugar to spike quickly along with increased insulin concentration and leads to an energy crash that can be detrimental to health [154,155]. One of the root causes of high glycaemic response in humans is due to the intake of breads, especially those loaded with carbohydrates and starches which are rapidly assimilated [115,156]. Realising the harmful effects of GI brought upon by bakery products, especially bread, more researchers have begun research on sourdough bread and revealed its advantages in terms of GI. In the International Tables of Glycaemic Index and Glycaemic Load values [157] and several scientific papers [38,39,40], it is reported that sourdough bread has lower GI than yeast bread. Whilst most gluten-free breads also tend to be high in GI due to low contents of fiber, micronutrients and protein, the sourdough is claimed to be one of the ingredients which reduces GI when being incorporated into gluten-free bread [158]. The study by Romão et al. [158] deduced that *Lb. plantarum*-based sourdough possesses lower GI than *W. cibaria*-based ones. Therefore, sourdough further benefits those suffering from celiac disease. By comparing different fermentation conditions, the decreases in GI in type-2 fermentation at 30 °C were significantly larger than type-1 fermentation at 25 °C in whole wheat bread samples [159]. The result was explained by higher level of lactic acid produced under these fermentation conditions. The presence of major fermentation and its end products of organic acids [160] such as lactic acid produced by LAB, like *Lb. sanfranciscensis*, *Streptococcus* spp. and *Leuconostoc* spp., in sourdough fermentation is effective in diminishing postprandial glycaemic and insulinemic responses, whereas acetic acid brings out a delayed gastric emptying rate [161,162,163,164,165]. In addition, the sourdough fermentation process also modifies carbohydrates’ molecular structure where starch availability is reduced under baking heat, thus allowing breakdown of sucrose to form exopolysaccharides which contributes to the rise in dietary fiber content [115]. This slows down the digestion of sourdough bread and eventually reduces the speed of sugar entering bloodstreams. The GI of bread is lowered regardless whether it is made from white, wholemeal or fiber-enriched flour [39,162,166,167,168,169,170]. Low GI diets have been shown to control blood glucose control better, increase insulin sensitivity, reduce serum cholesterol, reducing risk of developing type 2 diabetes, decrease risk of cardiovascular disease and improve weight control [41,42,43,44,45,46,47].

Sourdough which possesses low pH that ranged between 4.0 and 4.8 has the acidic environment that favours hydrolysis actions of enzyme phytase over yeast bread with 5.1–5.4 pH [171]. This is beneficial in terms of boosting the amount of nutrients because bread comprises decent amount of vitamins and minerals but phytic acid binds these important minerals and forms phytate which then renders these nutrients’ availability to our body [38,48]. The high amount of phytic acid in bread chelates cations to form an insoluble complex, phytate, which impairs the absorption of minerals and nutrients, such as iron, calcium, potassium, zinc into our body [171,172,173]. During sourdough fermentation, the starter culture, for instance the *Lb. plantarum* S18 and *L. mesenteroides* subsp. *mesenteroides* S50, breaks down phytic acid and reduces the phytate content by 62% in contrast with conventional yeast fermentation which only has phytate reduction of 38% [174]. De Angelis et al. [49] also studied this from a similar aspect and they concluded that solubility of minerals available in bread can be raised through addition of sourdough. The availability of these vitamins is adjusted through sourdough to regulate body metabolism and aid in the processing of food nutrients which help our body to feel energised and have a better mood throughout the day [50]. Besides increasing the availability of nutritional content, antioxidant and antihypertensive activities that protect body cells against free radicals which could reduce risks of diseases and cancers were also found in sourdough bread [51,52,53].

## 4. Conclusions

The dynamics and diversity of microbiomes in sourdough are known to give many benefits to bread and mankind. These are not only wholesome nutrients but also distinctive functions and benefits for health. This provides solutions to a large population whose lifestyle is inclined to eating bread as staple food, as conventional breads are laden with carbohydrate, highly starch-based and contain gluten which may have adverse effects on health. The sourdough microbes, particularly the lactic acid bacteria (LAB), give much benefits to our body’s gastrointestinal system. It improves our digestion system and increases nutrient absorption into our body. The sourdough fermentation process induced naturally by LAB and yeasts produces invertase enzymes which aid in digestion of short-chain carbohydrates present as non-digestible starch while the enzyme phytase will neutralise phytic acid present in grain-based products to free up the digestive enzymes of trypsin, pepsin and amylase to act more effectively in breaking down proteins, starches and fats in our body. The natural fermentation process in sourdough breadmaking helps patients with gastrointestinal disorders, e.g., irritable bowel syndrome (IBS) and also patients with celiac disease, e.g., gluten intolerance syndrome through alteration of carbohydrates and protein molecular structures, respectively. Consumption of sourdough bread allows better food digestion, promotes better nutritional uptake of minerals and vitamins and also improves gut health with the presence of LAB in the form of prebiotic food. From the point of view of artisanal baking, sourdough bread is always highly praised for its authentic and rustic look coupled with unique sensorial and eating properties. The organoleptic properties are much improved through enhanced flavour compounds and dough softening. The acidic environment resulting from the LAB fermentation gives the bread a longer shelf life. A profound understanding of each and every facet of sourdough fermentation affecting microbial ecology will help to improve the sourdough breadmaking process technology which is still quite a manual process. The application of sourdough into other food products such as noodles, desserts and snacks could be the next phase of exploration in order for sourdough benefits to reach greater heights.

## Figures and Tables

**Figure 1 microorganisms-09-01355-f001:**
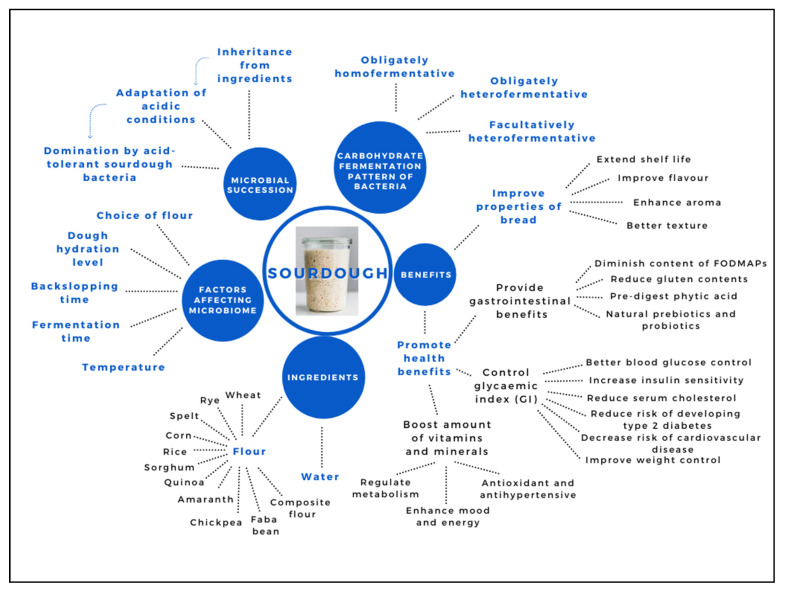
A mind map guide to sourdough.

**Table 1 microorganisms-09-01355-t001:** Benefits of sourdough as shown in Figure 1.

Benefits	References
Improved properties of bread	
(a) Extend shelf life	[12,18,22]
(b) Improve flavour	[12,17,18,23]
(c) Enhance aroma	[18,23,24,25]
(d) Better texture	[12,18,23]
Promote health benefits	
(a) Provide gastrointestinal benefits	
-Diminish content of FODMAPs	[26,27]
-Reduce gluten contents	[28]
-Pre-digest phytic acid	[29,30,31,32,33]
-Natural prebiotics and probiotics	[34,35,36,37]
(b) Control glycaemic index (GI)	[38,39,40]
-Better blood glucose control	[41,42,43]
-Increase insulin sensitivity	[41,42,44]
-Reduce serum cholesterol	[45]
-Reduce risk of developing type 2 diabetes	[41,42,46]
-Decrease risk of cardiovascular disease	[41,42,45,46,47]
-Improve weight control	[46]
(c) Boost amount of vitamins and minerals	[38,48,49]
-Regulate metabolism	[50]
-Enhance mood and energy	[50]
-Antioxidant and antihypertensive	[51,52,53]

**Table 2 microorganisms-09-01355-t002:** Summary of lactic acid bacteria (LAB) isolated from sourdough using different types of flour.

Flour	Dough (Prior to Fermentation)		Sourdough (After Fermentation)	
	LAB	References	LAB	References
Cereals				
Wheat			*Lb. acidophilus*	[14]
			*Lb. alimentarius*	
			*Lb. brevis*	
			*Lb. farciminis*	
			*Lb. fermentum*	
			*Lb. fructivorans*	
			*Lb. plantarum*	
			*Lb. sanfranciscensis *	
	*Aerococcus* spp.	[56]		
	*Lb. plantarum*		*Lb. plantarum*	[56]
	*Lb. sakei*		*Lb. sakei*	
	*Lb. sanfranciscensis*			
	*Lc. lactis*		*Lc. lactis*	
			*Leuconostoc* spp.	
	*P. pentosaceus*		*P. pentosaceus*	
	*Weissella* spp.		*Weissella* spp.	
	*Ec. hermanniensis*	[63]		
			*Lb. brantae*	[63]
			*Lb. lindneri*	
			*Lb. nodensis*	
			*Lb. plantarum*	
			*L. carnosum*	
			*L. kimchii*	
			*L. mesenteroides*	
			*P. argentinicus*	
			*P. pentosaceus*	
			*W. cibaria*	
			*W. confusa*	
			*W. salipiscis*	
	*Ec. lactis*	[64]		
			*Lb. brevis*	[64]
	*Lb. plantarum*		*Lb. plantarum*	
			*Lb. sanfranciscensis*	
			*Lc. lactis*	
	*L. citreum*		*L. citreum*	
	*P. pentosaceus*			
			*W. cibaria*	
			*Lb. fermentum*	[65]
			*Lb. johnsonii*	
			*Lb. plantarum*	
			*Lb. sakei*	
			*Lactococcus* spp.	
			*L. citreum*	
			*P. pentosaceus*	
	*A. viridans*	[76]		
			*Lb. curvatus*	[76]
	*Lb. fermentum*		*Lb. fermentum*	
			*Lb. plantarum*	
			*Lc. lactis*	
			*L. citreum*	
			*P. pentosaceus*	
			*W. cibaria*	
	*A. viridans*	[62]		
	*Ec. casseliflavus*			
	*Ec. durans*			
	*Ec. faecalis*		*Ec. cecorum*	[79]
	*Ec. faecium*		*Ec. faecium*	
	*Ec. mundtii*			
	*Lb. coryniformis*		*Lb. crustorum*	
	*Lb. graminis*		*Lb. delbrueckii*	
			*Lb. plantarum*	
			*Lb. sanfranciscensis*	
	*Lc. garvieae*		*Lc. garvieae*	
	*P. pentosaceus*			
			*Lb. brevis*	[75]
			*Lb. plantarum*	
			*Lb. acidophilus*	[69]
			*Lb. brevis*	
			*Lb. casei*	
			*Lb. crispatus*	
			*Lb. delbrueckii*	
			*Lb. fructivorans*	
			*Lb. kefirgranum*	
			*Lb. paracasei*	
			*Lb. pentosus*	
			*Lb. plantarum*	
			*Lb. rhamnosus*	
			*Lb. sakei*	
			*Lb. brevis*	[81]
			*Lb. curvatus*	
			*Lb. paralimentarius*	
			*Lb. plantarum*	
			*Lb. sakei*	
			*Lb. sanfranciscensis*	
			*Lb. zymae*	
			*Lc. lactis*	
			*L. citreum*	
			*L. mesenteroides*	
			*Lb. brevis*	[82]
			*Lb. sanfranciscensis*	
			*P. pentosaceus*	
			*W. cibaria*	
			*W. confusa*	
Rye			*Lb. kimchii*	[80]
			*Lb. paralimentarius*	
			*Lb. sanfranciscensis*	
			*Lb. spicheri*	
	*Aerococcus* spp.	[56]		
	*Lb. plantarum*		*Lb. plantarum*	[56]
	*Lb. sakei*		*Lb. sakei*	
	*Lc. lactis*		*Lc. lactis*	
	*P. pentosaceus*		*P. pentosaceus*	
	*Weissella* spp.		*Weissella* spp.	
	*Ec. casseliflavus*	[83]		
	*Ec. faecalis*			
	*Ec. mundtii*			
			*Lb. brevis*	[83]
			*Lb. fermentum*	
			*Lb. plantarum*	
	*L. citreum*			
			*P. pentosaceus*	
			*Lb. graminis*	[77]
			*Lb. plantarum*	
			*Lc. lactis* subsp. *cremoris*	
			*L. citreum*	
			*W. cibaria*	
			*W. confusa*	
			*Lb. brevis*	[84]
			*Lb. fermentum*	
			*Lb. amylovorus*	[85]
			*Lb. panis*	
			*Lb. reuteri*	
Spelt	*Ec. mundtii*	[76]		
			*Lb. brevis*	[76]
			*Lb. curvatus*	
			*Lb. paraplantarum*	
			*Lb. plantarum*	
			*Lb. rossiae*	
	*L. pseudomesenteroides*		*L. citreum*	
			*P. pentosaceus*	
Corn	*Lb. curvatus*	[62]	*Lb. acidophilus*	[86]
			*Lb. brevis*	
			*Lb. fermentum*	
			*Lb. plantarum*	
			*L. mesenteroides*	
			*L. dextranicum*	
			*P. acidilactici*	
			*Ec. saccharolyticus*	[87]
			*Lb. casei*	
			*Lb. delbrueckii*	
			*Lb. fermentum*	
			*Lb. plantarum*	
			*Strep. bovis*	
Rice	*Ec. faecium*	[62]		
			*Lb. brevis*	[88]
			*Lb. crustorum*	
			*Lb. harbinensis*	
			*Lb. pentosus*	
			*Lb. plantarum*	
			*L. pseudomesenteroides*	
Sorghum			*P. pentosaceus*	[89]
			*P. pentosaceus*	[59]
			*W. confusa*	
Pseudocereals				
Quinoa	*Ec. faecium*	[62]		
	*Ec. mundtii*			
	*Lc. garvieae*			
Amaranth	*Ec. faecium*	[62]		
	*P. pentosaceus*			
Non-conventional ingredients				
Faba bean			*Lb. sakei*	[90]
			*Lc. lactis*	
			*L. mesenteroides*	
			*P. pentosaceus*	
			*W. cibaria*	
			*W. koreensis*	
Chickpea	*Ec. faecium*	[62]		
	*Lb. graminis*			
Composite ingredients				
Wheat			*Lb. brevis*	[78]
Chickpea			*Lb. coryneformis*	
Lentil			*Lb. fermentum*	
Bean			*Lb. parabuchneri*	
			*Lb. paraplantarum*	
			*Lb. plantarum*	
			*Lb. pentosus*	
			*Lb. rossiae*	
			*Lb. sanfranciscensis*	
			*L. mesenteroides*	
			*W. cibaria*	
Apple			*Lb. plantarum*	[91]
Honey			*Lb. sakei*	
Wheat			*P. pentosaceus*	
Yogurt			*P. pentosaceus*	[91]
Wheat				
White grape			*Lb. brevis*	[91]
Wheat			*Lb. plantarum*	
			*Lb. sakei*	
			*P. pentosaceus*	
			*W. cibaria*	
Brazilian grape			*Lb. brevis*	[92]
Wheat			*Lb. casei*	
Rye			*Lb. delbrueckii*	
			*Lb. paracasei*	
			*Lb. rhamnosus*	

Abbreviations: *A., Aerococcus; Ec., Enterococcus; Lb., Lactobacillus; Lc., Lactococcus; P., Pediococcus; L., Leuconostoc; W., Weissella; Strep., Streptococcus*.

**Table 3 microorganisms-09-01355-t003:** Summary of yeasts isolated from sourdough using different types of flour.

Flour	Dough (Prior to Fermentation)		Sourdough (After Fermentation)	
	Yeast	References	Yeast	References
Cereals				
Wheat			*C. krusei*	[14]
			*P. anomala*	
			*S. cerevisiae*	
			*S. exiguus*	
			*C. humilis*	[56]
			*K. barnettii*	
	*S. bayanus*	[56]	*S. bayanus*	
	*S. cerevisiae*		*S. cerevisiae*	
			*W. anomalus*	
			*C. humilis*	[64]
			*K. barnettii*	
			*R. glacialis*	
			*S. bayanus*	
			*S. cerevisiae*	
			*C. humilis*	[74]
			*K. unispora*	
			*S. cerevisiae*	
			*S. uvarum*	
			*T. delbrueckii*	
			*C. humilis*	[79]
			*C. tropicalis*	
			*Cyberlindnera jadinii*	
			*S. cerevisiae*	
			*T. delbrueckii*	
			*W. anomalus*	
			*S. cerevisiae*	[95]
			*S. exiguus*	
			*C. humilis*	[81]
			*S. cerevisiae*	
			*P. fermentans*	
			*P. membranifaciens*	
			*W. anomalus*	
Rye			*C. humilis*	[56]
	*Cryptococcus* sp.	[56]		
			*K. barnettii*	
	*S. bayanus*		*S. bayanus*	
	*S. cerevisiae*		*S. cerevisiae*	
			*W. anomalus*	
			*C. humilis*	[74]
			*P. fermentans*	
			*S. cerevisiae*	
			*S. cerevisiae*	[84]
			*S. cerevisiae*	[85]
Sorghum			*S. cerevisiae*	[59]
			*S. cerevisiae*	[89]
Corn			*S. cerevisiae*	[86]
Rice			*S. cerevisiae*	[88]
Composite ingredients				
Brazilian grape			*C. famata*	[92]
Wheat			*C. guilliermondii*	
Rye			*C. pelliculosa*	
			*C. sphaerica*	
			*S. cerevisiae*	
Wheat			*P. kudriavzevii*	[96]
Lemon juice			*P. myanmarensis*	
			*S. cariocanus*	
Wheat			*Candida* spp.	[96]
Apple juice			*P. myanmarensis*	
			*S. cariocanus*	

Abbreviations: *C., Candida; P., Pichia; S., Saccharomyces; K., Kazachstania; W., Wickerhamomyces; R., Rhodotorula; T., Torulaspora*.

**Table 4 microorganisms-09-01355-t004:** Lactobacilli isolated from sourdoughs [1,102,105,106].

Obligately Homofermentative	Obligately Heterofermentative	Facultatively Heterofermentative
*Lb. acidophilus*	*Lb. acidifarinae*	*Lb. alimentarius*
*Lb. amylolyticus*	*Lb. brevis*	*Lb. casei*
*Lb. amylovorus*	*Lb. buchneri*	*Lb. curvatus*
*Lb. crispatus*	*Lb. cellobiosus*	*Lb. paralimentarius*
*Lb. delbrueckii* subsp. *bulgaricus*	*Lb. fermentum*	*Lb. plantarum*
*Lb. delbrueckii* subsp. *delbrueckii*	*Lb. fructivorans*	*Lb. pentosus*
*Lb. farciminis*	*Lb. frumenti*	*Lb. rhamnosus*
*Lb. helviticus*	*Lb. hilgardii*	
*Lb. johnsonii*	*Lb. panis*	
*Lb. lactis*	*Lb. pontis*	
*Lb. mindensis*	*Lb. reuteri*	
	*Lb. rossiae*	
	*Lb. sanfranciscensis*	
	*Lb. siliginis*	
	*Lb. spicheri*	
	*Lb. viridescens*	
	*Lb. zymae*	

**Table 5 microorganisms-09-01355-t005:** Selected biochemical reactions carried out by lactobacilli with different carbohydrate fermentation patterns [1,71,106].

Biochemical Reactions		Obligately Homofermentative	Obligately Heterofermentative	Facultatively Heterofermentative
Pentose fermentation		- ^1^	+ ^2^	+
CO_2_ released from	glucose	-	+	-
	gluconate	-	+	+
Carbohydrate digestion involving fructose-1,6-diphosphate aldolase		+	-	+
Carbohydrate digestion involving phosphoketolase		-	+	+

^1^ The group of lactic acid bacteria (LAB) does not carry out this biochemical reaction. ^2^ The group of LAB carries out this reaction.

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
