# Peer review of "Sourdough Microbiome Comparison and Benefits"

_microorganisms, 2021, doi:10.3390/microorganisms9071355_

Round 1

Reviewer 1 Report

Dear authors,

Thank you for the opportunity reading a thorough review of the current literature on sourdough. Your manuscript is overall a usefull material on the subject altough some major issues need to be addressed.

Major issue in any manuscript current;ly published in relation with LAB is the recently revised nomenclature on the genus Lactobacillus with the introduction of new genuses replacing a few of the species. 

I strongly believe that all current manuscripts should be revised accordingly  even more when they are review type and will be also considered as significant references for other scientists.

Your text also requires some editing in relation to the syntax and grammar used as well as the correct use of scientific terminology. Some but not all mistakes are as follows:

Line 15: "good bacteria" in scientific literature are referred as beneficial

Line 16: bacteria are not "consuming" water and flour! Rephrase please

Line 20: not all LAB are probiotics!

Line 61: although the significant interest of people in cooking and cooking shows as well as correctly pointed out in sourdough according to GOOGLE search that has been observed during the times of COVID19 pandemic it is not appropriate to refer to "obsessed" people.

Line 80 (and elsewhere): the authors use the term "constitution" apparently alternatively to populations. I think in clear direct phrasing the term constitution is inappropriate.

Line 105: ... climatic ...

Line 184: If your intiate the phrase with "Despite that..." there shouldn't be a new paragraph. Because the meaning can justify a new paragraph you need to rephrase.

Line 197:  Lb plantarum and Lb fermentum beside renaming according to recent literature on nomenclature they need to be italicised.

Line 221: heleveticus

Line 241: ... constitutions ... please reword.

Line 319: IBS is a common and potential disorder ... . What do you mean by "potential"? IBS is a common disorder.

Line 401: ... as staple food, ....

Line 414: ... to an environment for better food ... . This expression is not making sense. Please revise.

Line 420: ... longer shelf life.

Line 422:  .... technology.

Please revise your text with the aid of a native english speaker as some phrases were difficult to comprehend as well as some were also very long.

Author Response

Pls. see response attached.

Reviewer 2 Report

The manuscript titled “Sourdough Microbiome Comparison and Benefits” has been revised. The topic is interesting and well described, anyway some aspects need a deepening. I suggest the manuscript publication after major revision and an in-depth English mother tongue revision.

The authors can find the suggested correction in the attached file.

Regards

Author Response

Pls. see response attached.

Reviewer 3 Report

Although sourdough fermentation and bakery products are very important in our nutrition and health, this review is very disappointing. First of all the title promises interesting information but not fulfilled in the article. English language is very flamboyant and not suitable for any scientific journal, sometimes even advertisement like, certainly not scientifically objective. I really don’t think that this review offers new information. For a serious review, literature must be extensively researched. Overall disappointment.

Author Response

Pls. see response attached.

Round 2

Reviewer 2 Report

The authors made all suggested corrections. The manuscript can be accepted in the present form. 

Author Response

Thank you

Reviewer 3 Report

Dear authors,

improvement in language is noticeable. Again, some elementary errors are present (like subsp. not in italic...for any microbiologist this is major problem). My sincere opinion is that your review does not improves this field of research. For me, the final decision is rejecton. 

Author Response

We have changed “subsp.” In Table 2 to non-italic. We regret on overlooking this. As for comment that this paper does not improve this field of research, we are not agreeing because this review summarises important information about sourdough, which is very much trending eating lifestyle during this pandemic. It gives a good compilation of sourdough starters microorganisms from a wide and a variety of reliable resources. Whether it is a newcomer or an existing researcher, this review serves as a good reference and synthesis of information.